# The Use of the Foresight Methods in Developing an Algorithm for Conducting Qualitative Examination of the Research Activities Results on the Example of the Republic of Kazakhstan

**Rinat Zhanbayev [1,2]**, **Saule Sagintayeva [1]**, **Abildina Ainur [3]** and **Anton Nazarov [4],***

1   Rectorate, Almaty University of Power Engineering and Telecommunications Named after Gumarbek Daukeev, Almaty 050013, Kazakhstan; zhanbayevrinat@gmail.com (R.Z.); sagintayeva@mail.ru (S.S.)
2   Department of Scientific and Innovative Development, Republican Public Association «National Engineering Academy of the Republic of Kazakhstan», Almaty 050040, Kazakhstan
3   Management Department, Almaty University of Power Engineering and Telecommunications Named after Gumarbek Daukeev, Almaty 050013, Kazakhstan; editor@kisi.kz
4   Department of Business Informatics, Ural State University of Economics, 620144 Yekaterinburg, Russia
*   Correspondence: nazarovad@usue.ru; Tel.: +7-9222051888

**Abstract:** In modern conditions, it is interesting to study foresight as an effective tool for identifying new strategic scientific directions. Its purpose is to develop an algorithm for conducting qualitative expertise in the application of the foresight methods with the ability to integrate forecast estimates. Currently, the vast majority of research activities results do not contribute to the innovative development of the state. To solve this problem, it is necessary to ensure a stable systemic relationship between specific sectors of the economy and higher education. The algorithm is developed on the basis of a systematic approach to the foresight methods and the use of the methods of bibliometrics, scientometrics, patent analysis and forecasting. The results and conclusions of this study are: an algorithm has been developed for conducting qualitative examination of the results of scientific activities in order to increase its practical significance, in which the authors propose the foresight methods as the most optimal tool for choosing priority areas of science and technology. Putting this approach into practice will make it possible to increase the efficiency of the foresight methods by both reducing time costs, and rationally using monetary and human resources.

**Keywords:** foresight methods; research areas; expert forecast estimates; algorithm; research activities results

## 1. Introduction

The ability of the state to introduce innovations in rapidly developing industries and apply advanced technologies depends primarily on internal scientific, technical and industrial policies. Such a policy is the main factor supporting the innovative activity of economic entities and creating the conditions for new types of entrepreneurial activity. In this regard, research and development work and innovation as a tool for increasing productivity should be a priority for enterprises and the state, since they are the basis for the implementation of technical and economic progress as a significant multifaceted factor in increasing competitiveness and productivity [1].

As the current practice in Kazakhstan shows, the introduction of formal indicators designed to quantitatively reflect the effectiveness of scientific and technical activities immediately led to a change in the desirability of most of the scientific and technical community. Namely, after the h-index,

reflecting the performance of a particular scientist, was officially used, the publication activity of Kazakhstani scientists immediately increased. However, when making such decisions, the emphasis should be shifted to ensuring an increase in the level of scientific work, their scientific and applied significance. Otherwise, the task consists not so much in assessing the real state of affairs—it is known as well—as in creating the prerequisites for a qualitative leap, which is quite possible due to the availability of the necessary intellectual resources and effective tools, as well as the involvement of external intellectual resources (experts).

Scientists around the world are researching students' perceptions of e-learning platforms and tools. Recent research has shown that the use of electronic technologies and platforms is aimed at improving the educational process. Currently, Octavian Dospinescu finds the field of e-learning as the intersection of commercial, educational and technological interests [2].

Today, in most countries of the world (USA, Japan, Great Britain, France, Sweden, Russia, etc.), the foresight methodology has established itself as the most effective tool for choosing priorities in the field of science and technology. This methodology is used to predict all levels of scientific and technological development. Based on the foresight, medium- and long-term strategies for the development of the economy, science, and technology aimed at increasing its competitiveness are developed [3,4]. The main idea of the foresight is to determine the strategic directions of science, technology, economy, social sphere, etc., which in 15–20 years will become decisive for the development of an individual state and the entire world community [5–7].

Thus, foresight research allows you to develop integrated solutions that are at the junction of different scientific fields, types of economic activity and the competencies of officials and organizations. The involvement of experts for the development of scientific and technological forecasts is advisable, because, firstly, standard statistical methods and analysis of large data arrays alone do not allow to obtain medium and long-term forecast estimates and build scenarios for the development of the subject area, to identify qualitative trends in it and breakthrough innovations.

## 2. Conceptual Framework

As it was noted earlier, the foresight methodology is special and differs from others used in the field of foresight, it is open to absolutely everyone and does not impose restrictions on the participants in the process of creating the image of the future. This suggests that the result of the work should be a certain development concept, i.e., a vector that determines the direction of work for the long term, involving participants in the discussion of ideas in groups. These features allow you to seamlessly integrate foresight technology into the research process.

The study of a wide range of studies, one way or another related to the problems of this article, allowed us to group them in the following several areas:

-   The study of foresight as a system or concept for building the image of the future in the medium and long term, aimed at improving the quality of decisions currently being taken and coordinating joint actions;
-   Foresight is a systematic attempt to look into the future of science, technology, society and the economy to ensure the prosperity of society, economy and environment;
-   Foresight presents a system of methods for expert evaluation of strategic areas of socio-economic and innovative development, identifying technological breakthroughs that can affect the economy and society in the medium and long term.

In a number of publications, foresight is considered in the context of the knowledge economy [8,9], and the concept of strategic foresight [10–13]. Moreover, the knowledge economy is considered as an independent field of research, in the methodology of which various networks and the organization of a discussion platform are widely used.

Some scholars consider foresight as a concept of strategic foresight across time horizons, others as a concept of entrepreneurship.

Of greatest interest to us is the approach of the author Daisuke Kanama, who claims that technological foresight lies not only in planning a scientific, technical and innovative strategy, but also in creating a database as a training tool so that it can be used to formulate scientific technical and innovation strategy. The author, using bibliometric analysis, the Delphi method and scenario planning and technological planning, substantiates that the number of publications has increased, and the research methodology differs in each subject area [13].

These approaches are united by the fact that they are focused on building a future format through interaction with various actors in a certain environment [14].

It goes without saying that these studies have a certain angle of consideration of foresight issues, but taking into account the related nature of our research, the results of these studies were also of certain interest to us.

In the next group of publications, this approach is justified by the use of foresight, including at the local level. There, we note the emerging trend of considering the interaction between foresight and policymaking at the national and local levels. According to the authors, this approach is also well applicable for the formation of a national innovation system [15–17]. At the same time, according to some researchers (D. Eden, L. B. Methlie, G. E. Christensen), the theoretical aspects of this scientific problem are not sufficiently developed and systematized, and the scientific field is poorly organized [18].

Consideration of foresight by a number of authors (A. Meshkova, E. Ya. Moiseichev, 2016), as a research method that allows mobilizing available resources, proves its successful application in practice [19]. This allows us to conclude its functionality and applicability in the intended situations. Another group of studies is based on the analysis of state strategic documents, which allowed them to conduct predictive studies using the best world practice. Such studies were carried out taking into account expert assessments and demonstrate the validity of assessing development prospects using the foresight methodology [20,21].

However, in our opinion, the best result should be brought by the approach we applied, where the foresight methodology is complemented by a combination of forecast estimates. This approach is justified in this study.

Subscribing to the opinion that the transfer of technological knowledge from industrial research institutes is a complex and often doomed task, the authors, considering the foresight as a tool for determining future market needs and transforming them into a strategic scientific direction, are going to prove it is the foresight that is able to solve this problem.

To develop a view [21] on the impact of innovation policy on the direction of scientific research, the authors are going to justify the mechanism of encouraging firms to search for effective ways to obtain the optimal amount of research and development work. It is this view of the problem that we consider fundamental for our study.

In our opinion, the consideration of patents [22–25], only as a means of measuring innovation and technological change, is too limited. Although it should be noted that linking patents to indicators of economic activity opens up new possibilities for empirical patent analysis. This conclusion is justified by studies of a unique data set of 392 US patents obtained by well-known non-practicing entities from 1997 to 2006.

In the study, the authors develop the idea that it is not enough to be restricted to the effect of measuring innovation. It is necessary to determine the priority area of science. We develop this approach by applying criteria taken from the system of evaluation criteria for each research area. As a result, more perfect lists of research areas of research work are formed, each of which has a quantitative assessment based on the obtained values of the criteria, which will subsequently determine a number of priority ones.

This technique has already been used in the researches [26]. However, unlike our proposals, they emphasize the need for broad involvement, based on the expansion of dialogue with an emphasis on social issues. Of course, the wide involvement of all components is an important prerequisite for setting priorities. However, a guideline on expanding interaction balances the achievement of the

research goal, which is realized mainly due to focusing on social issues. The authors see the solution to the problem as an emphasis on mutual response in setting research priorities.

In our study, we substantiate that the formation of a list of promising research areas is carried out not only on the basis of patent analysis, but also on scientometric methods, bibliometrics, and also on the integration of predictive assessment. The algorithm under consideration also implements the possibility of expert forecast estimates.

The authors of the article, An Assessment Method for System Innovation and Transition (AMSIT) [27]—on a systematic approach to a retrospective study of the conditions for success or failure of innovative trajectories, we propose to expand the proposal by calculating the values of the evaluation criteria for each of the research areas when forming the initial lists of research areas. Of course, the authors of the method (AMSIT) reasonably believe that it helps to evaluate the initiatives of the innovation system that also face technological problems in the niche in which it operates, but also with the complex factors associated with the sociotechnical regime and the political and social landscape, where the niche is. Our methodology gives broader and more accurate possibility to more adequately cover not only research criteria, but also socio-economic ones, such as risk assessment, potential investor availability, applicability, prospects for entering the market and financing.

This is necessary to concretize the systems approach in the study and in the formulation of the question.

The study of the works of scientists on the problems of applying foresight as basic foresight methods revealed the following:

- There is a consistently high scientific interest in the problems of strategic forecasting in general;
- This scientific field is poorly studied and theoretical progress is not observed and, the most important thing is that there are no mechanisms for real practical application;
- We supplement the existing base of scientific research with the developed foresight methodology, namely, the methodology for assessing development prospects using the foresight methodology.

This study emphasizes the need for a thorough analysis of scientific and technological areas, for which an algorithm for identifying priority areas of science, technology and innovation is proposed. Moreover, the foresight is considered as a modern methodology, complemented by a combination of forecast estimates.

Thus, we can talk about filling the gap in the research devoted to foresight research, suggesting an integrated approach to this problem. We have developed an algorithm for making expert decisions in foresight studies to determine the strategic areas of science. Here, we see the advantage of using foresight as an element of influence on the future.

The concept of foresight has been studied quite deeply not only in developed countries, but also in Kazakhstan. The theoretical framework for Kazakhstani scientists in revealing the essence of foresight as a mechanism for determining the priorities of the formation of a knowledge society was the scientific achievements of R.S. Karenova, G. Schweizer, and B.D. Imanberdiyeva [28–30].

Summarizing the works of domestic and foreign scientists, we can conclude that the adequacy of the application of the foresight methodology is due to its advantages over other methods of expert evaluations, which include the following:

- Evaluate not only quantitative, but also qualitative information;
- Determine the weights of objects, processes and phenomena that are not amenable to quantitative measurement in the metric system of measures;
- Recreate an objective picture of the opinions of experts on the studied problem, which eliminates the averaging of their estimates (minimization of expert subjectivity);
- Establish a measure of the inconsistency of the judgments of each expert and thus determine the degree of reliability, confidence in the result [31].

The above features and advantages over other methods of expert evaluation constitute the peculiarity of foresight.

Thus, we consider the foresight methods is a set of tools that allow not to predict future problems, but to set a goal in the form of the desired expected result, to determine the necessary present condition. That is, it is an active forecast that includes elements of impact on the future.

## 3. Methods and Materials

The article uses the foresight methodology, which uses an integrated system of methods to determine the strategic areas of science by forming a list of technological areas. For these purposes, the article uses the following methods aimed at forming the initial list of the technological direction: methods of bibliometrics, scientometrics and patent analysis.

The method of counting the number of publications (bibliometric analysis), which involves quantitative assessment of bibliographic data, is used to systematize and compare the number of scientific publications in individual industries, which occupy a leading place in the structure of scientific knowledge [32].

Citation analysis in scientometrics is carried out by studying bibliographic references in publications of scientific periodical databases (Web of Science, SCOPUS, TR) in order to identify the citation of publications that form a certain area of science, the number of links (self-citation is excluded), and the total number of publications in a direction. According to the results of citation analysis, the most leading scientific areas are distinguished [33].

In patent analysis, statistical methods are used to process arrays of patent information, which include the analysis of dynamics curves of inventive activity in each scientific and technical field, and are based on the construction of cumulative patenting series characterized by an increase in the total number of patents related to this field [34]. We used these methods to form the initial list of research areas, their subsequent assessment at the fourth stage of the foresight methods, regardless of the subject area.

## 4. Methodology

Methods of bibliometry, scientifometry, patent analysis, method of observation and collection of facts, method of modeling were used.

Bibliometric analysis, or the method of calculating the number of publications, involves a quantitative assessment of the document flow and a quantitative analysis of scientific documents from different fields of knowledge. The fields of science and individual sections, which occupy a leading place in the structure of scientific knowledge according the number of scientific publications. distinguished on the results of bibliometric analysis.

Science measurement, or citation analysis, is carried out by studying bibliographic references in publications of scientific periodicals databases (Web of Science, SCOPUS, TR) in order to identify the citation of publications, forming a certain direction of science, the number of references (self-citation is excluded), the total number publications by direction.

Patent analysis uses statistical methods for processing patent information arrays including the analysis the dynamics of inventive activity's curves for each scientific and technical direction; and the result of analyses includes the construction of cumulative patent series characterized by an increase of the total number of patents related to this area. Our methodology allows us to cover not only research criteria, but also socio-economic factors, such as risk assessment, the presence of a potential investor, applicability, market access prospects and financing.

It is necessary to specify the systemic approach in the study and in raising the question.

The adequacy of the application of the foresight methodology is due to its advantages over other methods of expert assessments, giving the following possibilities:

- Evaluating not only quantitative but also qualitative information;
- Determining the weight coefficients of objects, processes and phenomena that cannot be quantified in the metric system of measures;

-   Recreating an objective picture of experts' opinions on the problem being studied, which excludes the averaging of their assessments (minimization of expert subjectivity);
-   Establishing a measure of contradictory judgments of each expert and thus determine the degree of reliability, trust in the obtained result.

## 5. Results

In modern business conditions, it is becoming increasingly difficult to conduct research and development works simultaneously on a full range of scientific areas. To ensure the ongoing development of the economy, it is necessary to adequately and timely identify the priorities for future development using traditional and innovative approaches, including modern methods of analyzing big data and expert information. Foresight research is aimed at determining the prospects for socio-economic, scientific, technological and innovative development, as a rule, with a horizon of 10 years or more. Almost always, foresight involves the synthesis of practical techniques necessary to achieve the desired analytical depth of the results obtained during the study. Therefore, considerable attention is paid to the multi-level work of experts in identifying scenarios and development priorities for the subject area in the long term. The foresight process includes the elements of interaction and consideration of the views of leading experts in the subject area, including decision makers. The main advantages of this approach are the involvement of various parties in the process of developing managerial decisions, starting from the early stages of forecasts development, as well as the organization of interaction between representatives of various scientific disciplines and areas of activity in the process of research.

In the course of research, we developed an algorithm for conducting a qualitative assessment of the expertise, the structure of which is shown in Figure 1. The algorithm consists of five main parts (steps): setting up a foresight research model; the formation of expert panels; formation of the initial list of research areas; assessment and refinement of the initial list of research areas; coordination and approval of priority areas (Figure 1).

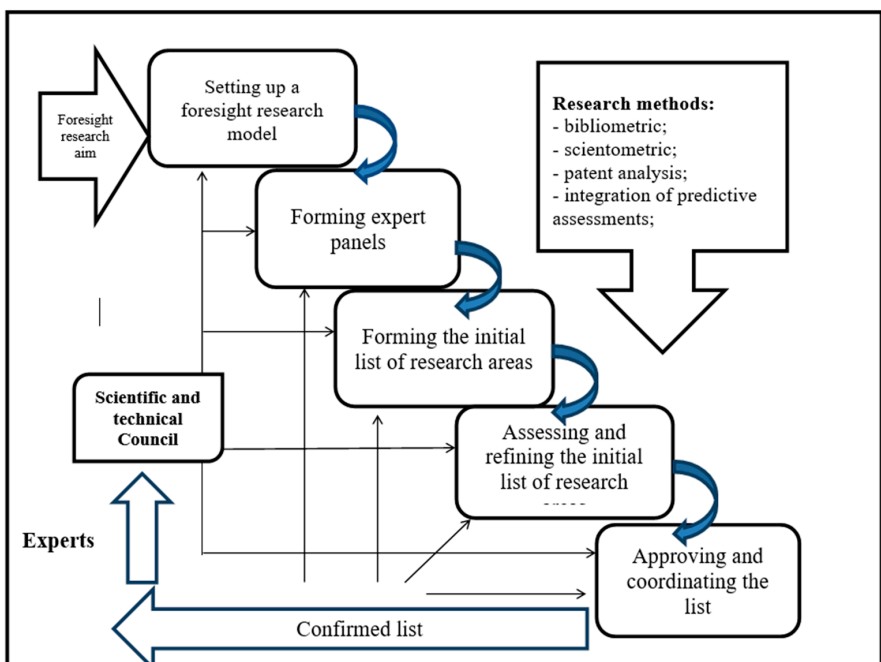

**Figure 1.** Algorithm for qualitative expertise in conducting foresight studies.

(1)　At the first stage, it is necessary to choose the type and main goal of foresight research. After that, the foresight research model is set up, that is, the content and sequence of the stages of the foresight research are determined, as well as a set of methods for each of the stages.

(2) At the second stage, operations are performed such as calculating the required number of research experts and the generalized indicator of the competence level of each expert based on coefficients reflecting both the level of professional training and the personal qualities of the expert.

The composition of experts participating in the assessment is a key parameter that determines the quality of the future forecast. Thus, to conduct a quality examination, it is necessary to solve the following tasks:

(1) Assess the competence level of experts;
(2) Determine the size of the expert group;
(3) Create a final list of experts participating in the examination.

$$K_i = \frac{1}{4} \sum_{J=1}^{4} K_{ij} \tag{1}$$

we denote by Q the set of experts, then to assess the competence level of each *i* expert ($i = 1, \ldots, m$) we use the generalized indicator of the competence level ($K_i$) given in [35], which takes into account both professional activity and personal qualities of experts:

Where $K_{i_1}$ is the coefficient reflecting the level of professional training and awareness of the *i* expert (takes into account the qualification levels "doctor of science", "candidate of science", etc. and is measured in points of $0.5 \le K_{i_1} \le 1$);

$K_{i_2}$—the coefficient reflecting the level of basic argumentation of the *i* expert when making his decision (takes into account factors such as intuition, production experience, theoretical analysis, etc. and is measured in points of ($0.5 \le K_{i_2} \le 1$);

$K_{i_3}$—the coefficient reflecting the personal qualities of the *i* expert, and calculated on the basis of self-esteem ($0.5 \le K_{i_3} \le 1$):

$$K_{i_3} = \frac{1}{n} \sum_{J=1}^{n} K_{i_3 j} \tag{2}$$

where $K_{i_3 j}$ is the coefficient reflecting the self-esteem of the *i* expert on his *j* personal quality; *n* is the number of personal qualities of the expert; (characterized by his behavior and relations with the verifiable) [36,37].

$K_{i_4}$ is the coefficient reflecting the personal qualities of the *i* expert, and calculated by colleagues experts ($0.5 \le K_{i_4} \le 1$):

$$K_{i_4} = \frac{1}{n \times m} \sum_{i=1}^{m} \sum_{J=1}^{n} K_{i_4 j} \tag{3}$$

$K_{i_4 j}$ is the coefficient given by the *i* expert about the *j* personal quality of the *i* expert; *n*—the number of personal qualities of the expert; and *m*—the number of the experts participating in the assessment of the *i* expert.

As a criterion for assessing the required number of experts, we use the following formula:

$$N_{min} = 0.5 \left( \frac{3}{\varepsilon} + 5 \right), \tag{4}$$

where *N* min is the minimum number of experts; $\varepsilon$—the parameter setting the minimum level of examination error ($0 < \varepsilon \le 1$).

With a permissible error of expert analysis of 5% ($\varepsilon = 0.05$) the number of experts should be at least 32.

According to [38] the required number of experts for group assessment should be at least 7–9 people, therefore, the number of experts involved in forecasting is within $7 \le N \le 32$.

In order to get a final list of all experts who have passed certification, they are ranked according to the level of competence (the value of the generalized indicator *Ki*) and in accordance with relation (4) a list of experts participating in the examination is formed.

In the framework of the third stage, the initial list of research areas is formed, for its subsequent assessment, and the values of the criteria for their assessment are calculated.

It is necessary to divide the task posed above into a number of subtasks:

(a) Create an initial list of research areas, for their subsequent evaluation at the fourth stage of the foresight methods;
(b) Calculate the values of quantitative assessment criteria for each of the areas;
(c) Get the values of the qualitative evaluation criteria for each of the areas.

To solve the first of the set sub-problems, using the methods of bibliometrics, scientometrics and patent analysis, it is necessary to form an initial list of research areas. At the same time, it is proposed to use methods for integrating forecast estimates [37], which will improve the accuracy of the selection of research areas for the formation of their initial list.

To solve the two remaining problems, it is necessary to calculate the values of the criteria from the system of evaluation criteria for each research area.

As a result, more advanced lists of research areas are formed, each of which has a quantitative assessment based on the obtained values of the criteria for their assessment, which will subsequently determine the number of priority ones. The list of research areas is formed by the methods of scientometrics, bibliometrics, patent analysis, as well as the integration of forecast estimates.

It should be noted that at present there is no single foresight model, and each country adapts it to its own conditions taking into account national interests, using different forecasting techniques. In our opinion, the use of patent analysis is acceptable to determine the priority area for the development of science in Kazakhstan.

In patent analysis, in order to obtain quantitative characteristics of the development of certain areas of science and technology, statistical methods are used to process arrays of patent information [34] (b).

The analysis showed that the development of Kazakhstani science is concentrated in the following areas: natural sciences and engineering developments and technologies, more than 60% of researchers are involved in them (Figure 2).

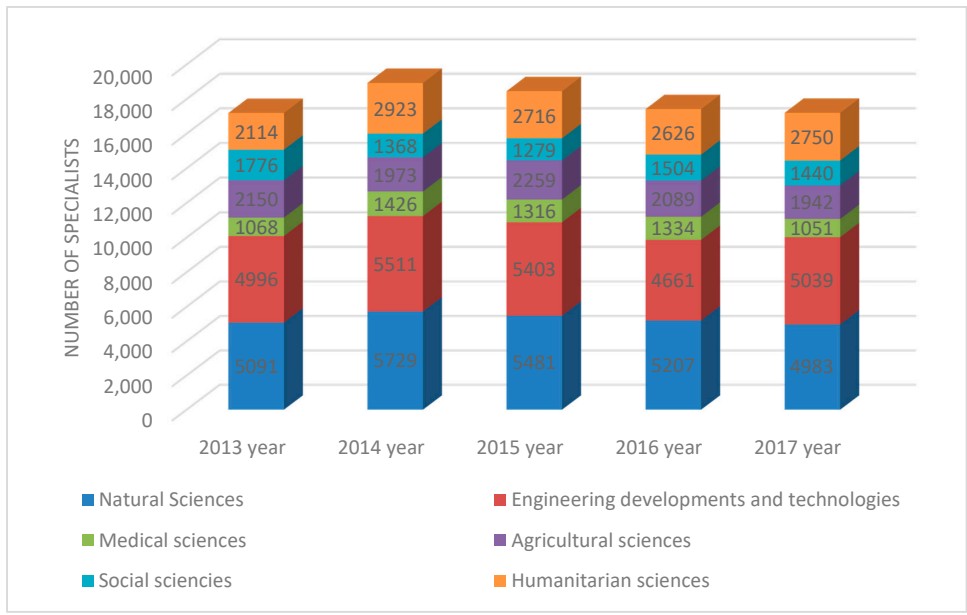

**Figure 2.** Dynamics of the number of research specialists who performed research and development work by branches of science.

As a part of the patent analysis, we will analyze the patents issued in Kazakhstan. The number of patents granted is 869, while 36,454 patents have been issued in Russia, 327,000 in China, 324,760 in the USA, and 318,364 in Japan (Table 1).

**Table 1.** The number of patents granted in Kazakhstan for 2013–2017.

| Indicator | 2013 | 2014 | 2015 | 2016 | 2017 |
|---|---|---|---|---|---|
| Granted protective documents for inventions | 1500 | 1504 | 1504 | 1011 | 869 |
| Granted utility models patents | 163 | 165 | 166 | 577 | 591 |
| Granted protective documents for design inventions | 280 | 282 | 282 | 182 | 129 |

The distribution of protection documents issued in 2017 for inventions by sections of the International Patent Classification, which covers all areas of knowledge whose objects may be protected by protection documents, is presented in the table (Table 2), the data of which indicate the prevalence of protection in Sections A «satisfaction of vital needs of person» (30%) and C «chemistry and metallurgy» (24.5%).

**Table 2.** Distribution of granted patents for inventions by sections of the International Patent Classification (2017).

| Sections of IPC | Number of Granted Patents | | Total |
|---|---|---|---|
| | For Inventions | For Utility Models | |
| A Satisfaction of vital needs of person | 269 | 159 | 428 |
| B Different technological processes | 116 | 98 | 214 |
| C Chemistry; metallurgy | 230 | 129 | 359 |
| D Textile; paper | 2 | 2 | 4 |
| E Construction, mining | 72 | 72 | 144 |
| F Mechanics; lighting; heating | 71 | 58 | 129 |
| G Physics | 72 | 49 | 121 |
| H Electricity | 37 | 24 | 61 |
| Total | 869 | 591 | 1460 |

Despite the positive dynamics, there is a low effectiveness of scientific research. Therefore, for 17 thousand scientists there are 0.07 applications for inventions. In the country, the number of patent applications per 1 million population is 0.0008 (for comparison, in Russia—195.9; Germany—582.6; Great Britain—289.7; USA—741.8; Korea—2591.5; and Japan—2720.7).

The method of calculating the number of publications (bibliometric analysis) is as follows: a quantitative assessment of the document flow is organized within the framework of one of the accepted classifications, i.e., the analysis of the number of scientific documents from different areas of knowledge that have been reviewed is carried out. Thus, areas of science and individual sections, which occupy a leading place in the structure of scientific knowledge by the number of scientific publications are distinguished. Then, the number of publications by individual branches is compared in order to single out the «leading» branches of knowledge (as a percentage of the total number of publications referenced for a given period).

As Figure 3 shows, the majority of scientific publications were published in the field of physics and astronomy—485, materials science—426, agrarian and biological sciences—291, chemistry—277, mathematics—178 and the smallest numerical publications were in the field of Earth science—85.

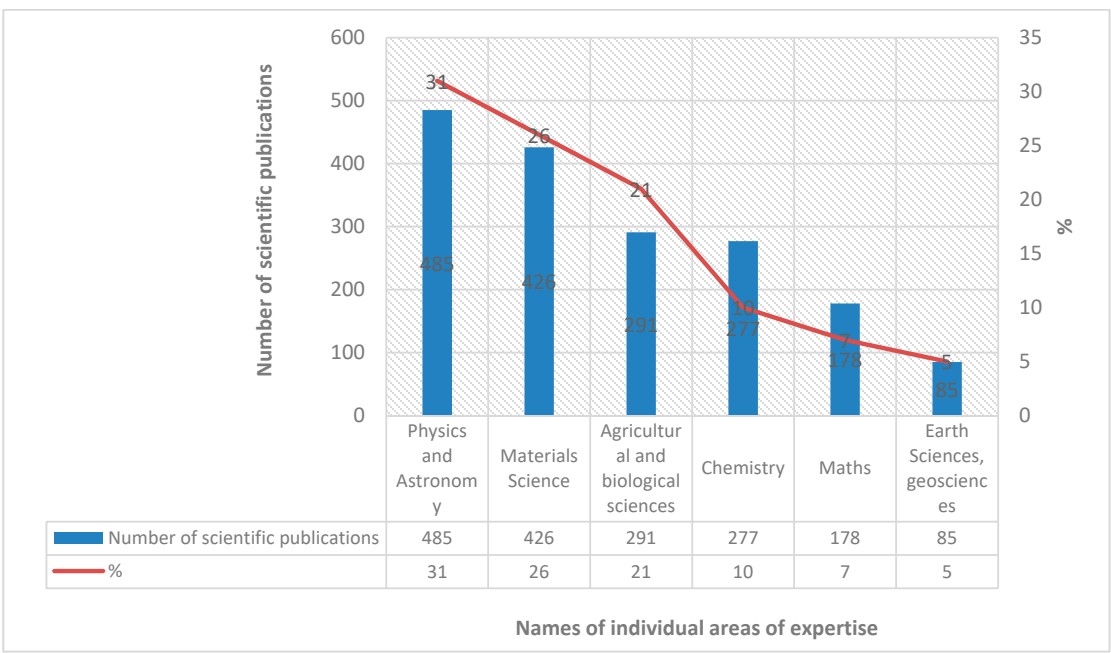

**Figure 3.** Bibliographic indicators of Kazakhstan in selected areas of knowledge for 2017.

Kazakhstan is leading in terms of the growth rate of scientists' publication activity among OECD (Organization for Economic Co-operation and Development) countries. Thus, in 2019, 4747 publications of Kazakh authors were indexed in the Scopus database, and 3590 publications were indexed in the Web of Science database, which is 8.2 and 8.4 times higher than in 2011. At the same time, the number of authors who published at least one work indexed in the Web of Science database in 2019 increased 6.2 times compared to 2011 (Figure 4).

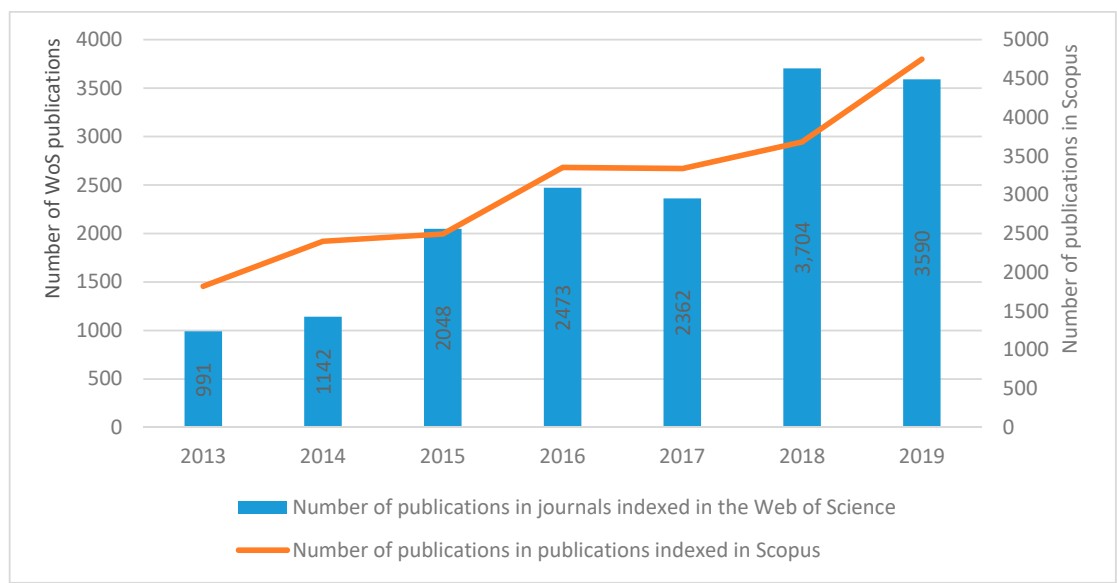

**Figure 4.** The number of Kazakhstani publications in foreign rating publications indexed in Web of Science and Scopus.

In Kazakhstan, the number of scientific publications of Kazakhstani scientists is 2362, while in Russia—72,085, in Ukraine—10,986. If we talk about scientific works in foreign indexed publications, their share in the world stream is 0.08% (Web of Science (Thomson Reuters, Toronto, ON, Canada)) and 0.1% (Scopus (Elsevier, Amsterdam, The Netherlands)). Based on the analysis of citation of Kazakhstani

publications in international peer-reviewed journals, the most leading scientific areas are mathematics, physics and astronomy.

According to the algorithm, as a result of solving the first of the set sub-problems, we received a ranked list of research areas, the final composition of which should be clarified and approved by the Scientific and Technical Council, as prescribed by the foresight methods, for further calculation of the values of the criteria for their assessment.

The next step in the procedure of generating the initial lists of research areas is to solve the second and third sub-tasks, namely, to calculate the values of the evaluation criteria for each of the research areas, which are provided by the foresight research methodology (Table 3).

**Table 3.** System of criteria for assessing research areas.

| Research Areas Assessment Criteria | |
|---|---|
| **Socio-Economic Criteria:** | **Research Criteria:** |
| - risk assessment; | - availability of a scientific leader; |
| - availability of a potential investor; | - availability of production capacity; |
| - applicability; | - availability of scientific specialists; |
| - prospects for entering the market; | - effectiveness of research activities; |
| - financing. | - availabilityof an experimental production base. |

Currently, the governments with developed market economies are pursuing policies of creating a so-called "new economy" or "knowledge-based economy." The active formation of the new economy has been carried out over the past twenty years with global information networks, communications and the Internet. The strategic development of the national economy of various countries, including Kazakhstan, is primarily inextricably linked with the formation of new economic relations. The authors understand the economics of knowledge to be the system of socio-economic relations of the innovative, digital type, based on the priority of intellectual human capital, the intensive development of digital technologies, knowledge-intensive production, high-technology industries and continuing professional education.

Criteria, according to their values, are divided into quantitative and qualitative, obtained by expert evaluation.

Quantitative criteria (expressed by quantitative values) include—financing; availability of scientific specialists; effectiveness of research activities; availability of production facilities and availability of an experimental and production base.

To find the values of the criterion "Financing" ($K_f$), the share of capital costs in the internal costs of research, development and purchase of equipment is determined [38].

$$K_f = \frac{Z_{res} + Z_{cap}}{Z_{intern}} * 100\% \tag{5}$$

where $Z_{res}$—research and development capital expenditures; $Z_{cap}$—capital and current expenses for the purchase of equipment; $Z_{intern}$—internal research and development costs.

The criterion of «availability of scientific specialists» ($Ks$) is the specific gravity of the number of highly qualified specialists in the total number of scientists [39]:

$$K_{ss} = \frac{K_{qs}}{K_{tot}} * 100\% \tag{6}$$

where $K_{qs}$—the number of highly qualified specialists; $K_{tot}$—the total number of scientists.

Specialists of the higher classification include doctors and candidates of sciences, and their numbers are calculated taking into account weight coefficients of 0.8 and 0.5 for doctors and candidates of sciences, respectively:

$$K_{hq} = 0.8 * K_{doc} + 0.5 * K_{cand} \tag{7}$$

The criteria "availability of an experimental and production base" (Kex) and "availability of production facilities" (Kpf) are determined by analyzing accounting and statistical reporting documents (information on the availability of production capacities, balance sheet, balance of production capacities, etc.) of a university of a particular research area. These two criteria take the value 1 if the availability is confirmed and 0 if this is not the case [40,41].

The criterion of «research activity efficiency» ($K_{rae}$) implies a total calculation of development indicators for a particular research area, such as—the total number of valid university patents ($K_{vup}$); the number of filed patent applications and applications for utility models (for the last year) ($K_{pat}$); and the number of acquired rights to patents and patent licenses ($K_{rpat}$):

$$K_{rae} = K_{pat} + K_{vup} + K_{rpt} \tag{8}$$

Qualitative criteria or criteria—the values of which are obtained by expert assessment, include applicability, availability of a potential investor, prospects for entering the world market, and availability of a scientific leader. To determine the values for each of the qualitative criteria, experts need to fill out a questionnaire obtained by merging templates from several questionnaires in the foresight research methodology.

To process group expert assessments, it is necessary to check the degree of expert opinions consistency [42], and then determine the generalized (aggregated) group assessment [43] for each research area on each of the criteria.

As indicators of the degree of expert opinions consistency, the coefficient of variation that characterizes the relative dispersion of the result is used:

$$V_j = \frac{\sqrt{\frac{\sum_{i=1}^{m_j}(x_{ij} - \overline{x}_j)^2}{m_j - 1}}}{\overline{x}_j} * 100\% \tag{9}$$

where $V_j$ is the coefficient of variation of the estimates for $j$ research area; $m_j$—number of experts evaluating the $j$ research area; $x_{ij}$—assessment in points by the $i$ expert of $j$ research area; and $\overline{x}_j$ is the average statistical value of the research area assessment value in points:

$$\overline{x}_j = \frac{\sum_{i=1}^{m_j} x_{ij}}{m_j} \tag{10}$$

The smaller the value of the coefficient of variation is, the more consistent is the opinion of experts. If there is no agreement among experts, the second assessment is carried out. Experts are sent an additional information about the subject of the expertise and they as a rule adjust their grades. The adjusted information is again sent to the analytic group to verify consistency.

To calculate the aggregated group assessment, we use the method of average scores. Taking into account the weight coefficient of the experts, the group estimate of $j$ research area is calculated as the weighted average:

$$x_j^{c_B} = \sum_{i=1}^{m_j}(K_i * x_{ij}) \tag{11}$$

where $K_i$ is weighting coefficients of experts competence; $m_j$—number of experts evaluating the $j$ research area; $x_{ij}$—assessment in points by the $i$ expert of $j$ research area.

At the fourth stage of the foresight methods, it is necessary to select the priority areas of research and development by evaluating among themselves the research areas obtained at the previous stage. The stage of assessment and refinement of the list of research areas is to evaluate research areas by the values of their criteria.

The research area is evaluated using the Pareto optimality principle and the t-ranking method (to narrow the Pareto domain). As a result, research areas, whose vector estimates make up the Pareto set, are priority areas.

Let us consider each of the methods in more detail.

The mathematical model of the task of choosing the most priority areas of research work can be represented as:

$$B_f = < X, f_1, f_2, \ldots, f_m >$$

where $X$ is the set of research areas; $f_j$—the numerical function defined on the set $X$, and $f_j(x)$ is the estimate of the research areas $x \in X$ by the $j$ evaluation criterion ($j = \overline{1, m}$).

The goal of solving the problem of choosing the most priority areas of research is to obtain areas that have the highest possible scores for each criterion, i.e., in highlighting the Pareto set [42]. All criteria functions $f_j$ reflect the usefulness of research areas $x \in X$ from the standpoint of various assessment criteria and should be commensurable, i.e., the values of each criterion function vary within the same limits [a, b]:

$$\forall x \in X \div 0 \leq a \leq f_j(x) \leq b, j = \overline{1, m}$$

At the same time, the least preferred by any of the particular criteria $f_j(x)$ research area will receive an assessment $a$, and the most preferable-assessment $b$ ($a = 0$, $b = 1$).

The above mentioned numerical functions $f_j(x)$ ($j = \overline{1, m}$) form a vector criterion $f = (f_1(x), f_2(x), \ldots, f_m(x)) \in R^m$, where $R^m$—space $m$—dimensional vectors.

For any research area $x \in X$ the set of its estimates by all criteria, i.e., the set $(f_1(x), f_2(x), \ldots, f_m(x))$ is the vector estimate of research areas $x$. All possible vector estimates form the set of possible estimates $Y = f(x) = \{y \in R^m | y = f(x)$ for some $x \in X\}$.

The Pareto dominance ratio is defined as follows: research areas $x_i$ dominates the Pareto research area $x_j$, if the vector estimate $f(x_i) = (f_1(x_i) \ldots, f_m(x_i))$ research area $x_i$ dominates the Pareto vector estimate $f(x_j) = (f_1(x_j) \ldots, f_m(x_j))$ research area $x_j$, i.e., if there is inequality $f(x_i) \geq f(x_j)$, and therefore $x_i \geq x_j$. Substantially, the Pareto dominance condition means that research area $x_i$ is no worse than research area $x_j$ according to any of the considered criteria, and at least one of these criteria $x_i$ is better than $x_j$.

Therefore, according to the values of the criterion functions ща research area, we obtain their vector estimates. To find the set of Pareto-optimal vectors P (Y), we compare them with each other according to the rule described above. If the obtained pairs turn out to be incomparable with respect to the Pareto, then the task is to narrow the initial set using the t-ranking method, in order to select several research areas as the final result.

The task is to select several research areas as the final result by narrowing the Pareto set. One of these methods is the t-ranking method [44], which uses the ordinal information of the decision maker (DM) about the relative significance of the criteria.

As the initial information for the t-ranking method, the set of $S$ statements of the decision maker about the relative importance of the criteria for evaluating the form is taken:

$$S = \left\{ f_k = f_j; \ldots; f_q > f_p \right\}, \tag{12}$$

which needs to be expanded by adding new transitive statements that are consequences of existing ones.

Taking into account the obtained set (12), when comparing two vector research area estimates, a preference relation is constructed by the t-ranking method:

$$\ni \left( (x_z)^t \right) > f_1(x_w) \longleftrightarrow \left[ \exists f''^{(x_w)} \in f''^{(x_w)} I \div f(x_w) > f''^{(x_w)} \right], \tag{13}$$

$$f(x_z) > f(x_w) \longleftrightarrow \forall j \in [1 \div m] \div f_j(x_z) \geq f_j(x_w)$$

where $f(x_z), f(x_w)$ research area *vectorratings*.

$(f(x_Z) = (f_1(x_Z), \ldots, f_m(x_z)); f(x_w) = (f_1(x_w), \ldots, f_m(x_w))); f(x_w)I$—a set of $f(x_w)$—improved vectors $(f_k = f_j; f_q > f_p)$.

Thus, the initial data of the problem are the set (12) of ordinal information on the relative importance of the criteria and the set of Pareto-incomparable vectors. A pair of vectors $f(x_i)$ and $f(x_j)$ is selected to compare their vector estimates. We fix the vector $f(x_i)$ and by the vector $f(x_j)$ we obtain the sets of improved vectors $f(x_j)$ I according to (13). After the transformations, we get Pareto comparable vectors. As a result, we get a lot of Pareto-optimal vectors, and as a result, the priority areas of research work [45].

The fifth stage serves to coordinate and approve the priority areas of research obtained at the previous stage. At this stage, a document is compiled, which is a final report and which includes the results of the previous stages, namely: a list of the experts participating in the expertise, a list of initial research areas with values for each criterion for their assessment, and a ranked list of priority research areas research work.

In case of positive approval, a document is formed with a list of approved priority areas of research. Otherwise, according to the rules of the foresight methods, the input information for the algorithm is refined, if necessary, adjustments are made, and a document with recommendations for restudy is formed.

Within the framework of the concept of a comprehensive evaluation of foresight projects, the coordination and approval of the list of thematic areas will be implemented in accordance with the approach described in [46]. In accordance with this approach, a special scientific environment is synthesized for the implementation of foresight methods, in which all participants in the process are immersed.

## 6. Discussion

Labor costs and time spent on research work represent a well-defined resource that can be used for the innovative development of the state, including Kazakhstan. Currently, this resource is mainly aimed at fulfilling the formal requirements for research work; their results in the vast majority of cases do not find real practical application and do not contribute to the development of the national economy in terms of the commercialization of the results of scientific and technical activities.

Studies have shown that the results of studies on this problem, which are widely represented in the scientific literature, are not able to solve the raised question of increasing the contribution of research to the innovative development of the state.

We believe that in order to solve the above mentioned problem, it is necessary to provide a stable systemic connection between specific sectors of the economy and higher education. Scientific and technical problems to which higher school directs its efforts should be determined primarily by the demands of the economy and society, and not by the response rate factor, when the same scientific or technical problem has been exploited by a specific research institute or university for decades, irrespectively of the possibility (or lack of possibility) of the commercialization of the scientific and technical activities results. This problem is essentially a systemic one, and not only in Kazakhstan.

According to the tradition that has developed in the post-Soviet space, the planning of scientific activity comes from the scientific interests of a particular university, department, or specific scientist (development is created first, and only then opportunities are found for its promotion on the market). In nowadays conditions, it is necessary to switch to a different scheme, when the choice of the direction of scientific and technical activity is preceded by a marketing and economic study of the issue about the appropriateness of carrying out relevant research.

To solve this problem, a tool is proposed that is used for the exchange of information between the business communities, industrial organizations and other domestic structures that are able to select and adequately formulate real tasks that need to be addressed for the accelerated innovative development of the national economy.

This study complements the theory and practice of foresight as a forecasting technology, namely:

1.  Allows us to evaluate, compare and summarize the results of a large number of different strategic studies on the issue;
2.  Helps to identify growth points of the economy by applying topical foresight techniques to assess potentially promising, commercially viable activities in the business environment;
3.  Provides systematic constructive interaction between the representatives of science (undergraduates, doctoral students, professors) and the expert community to identify and develop promising areas of scientific and technical activity;
4.  The results of this study adjust the progress of research work in terms of the commercialization of scientific and technological activities;
5.  Reduces labor costs, time spent in identifying and considering advanced topics of scientific work of undergraduates, doctoral students, professors and provides a stable systemic link between specific sectors of the economy and higher education.

The study of the works of foreign scientists showed that despite the increased scientific interest in forecasting, this scientific field is poorly studied and needs further research. We have developed the methodology for assessing development prospects using foresight technology.

The proposed approach simultaneously provides the solution of the following urgent problems:

-   Improvement of the quality of scientific and technical work due to continuous effective control over the progress of their implementation;
-   Creation of a discussion platform designed to discuss the progress of these works and identification of weaknesses based on assessing in an open discussion.

It is appropriate to emphasize that the capabilities of monitoring schemes for the progress of scientific and technical work traditionally used in post-Soviet universities and research institutes are limited, primarily due to the subjectivity factor, which is inevitably present in assessments with a small number of experts involved. The transition to foresight-oriented forms will eliminate this factor. Thus, on a national scale, the results of the study can ensure the creation of new forms of stimulation of scientific and technological activities aimed at mobilizing such a resource as labor costs of research institutes workers. On an international scale, the article represents a unique experiment in the field of institutional economics, aimed at the directed creation of an institution that ensures the identification of the most promising scientific areas in the mode of self-organization.

The intellectual environment in Kazakhstan is fragmented and overcoming this state of affairs, at a minimum, requires a reorientation of a significant number of researchers to new scientific directions for them. The traditional methods currently used in the country (through material incentives) are very costly.

We believe that the results of the study are more than significant, since the proposed approach is able to solve the most important scientific and social problem that remains unresolved so far—providing a systematic connection between science and business based on a systematic approach to the selection of critical scientific areas that ensure industrial development of economy of any state. The same factor determines the social effect—the creation of platforms for a systematic exchange of views and a critical analysis of existing research areas.

To achieve these goals, the paper proposes the modernization of foresight methods, which help to identify the most promising areas of scientific and technological development. Modernization will be carried out on the basis of the application of the expert decision-making algorithm, which consists of five main parts (steps): setting up the foresight research model; formation of expert panels; formation of the initial list of research areas; assessment and refinement of the initial list of research areas; coordination and approval of priority areas.

The application of the algorithm developed by the authors is directed to the most efficient use of the available intellectual potential in order to solve the real urgent problems facing the national economy, the commercialization of the results of scientific and technical activities.

The project team piloted the development of Forsite information technology in the educational process of undergraduates and doctoral students, and worked out administrative procedures involving experts and students. The web application (information system) was introduced in the NAO "Almaty University of Energy and Communications". We consider our algorithm proposal to be efficient in the Commonwealth of Independent States (CIS) and in European Union (EU). On an international scale, the project is a unique experiment in the field of institutional economy, aimed at creating an institution that identifies the most promising scientific directions in the self-organization mode.

Thus, we can talk about a new direction in research devoted to forecasting the potential areas of scientific and technological activity. We designate this direction as an unexplored one and that is in demand both in the global scientific environment and in practical implementation.

## 7. Conclusions

The studies have shown that using the foresight method to improve the algorithm for conducting qualitative expertise in order to increase the effectiveness of research activities allows us to realize a comprehensive approach to the problem of systematic constructive interaction between undergraduates, doctoral students, professors and the expert community to identify and develop promising areas of scientific and technical activities.

The analysis of practical experience and theoretical developments indicates the absence of an algorithm for conducting qualitative expertise in the implementation of the foresight methods with the ability to integrate predictive assessment. In the framework of this study, various approaches were systematized and generalized, its most important elements and relevant criteria were identified, which, in our opinion, will help to increase the effectiveness of research activities.

The proposed algorithm realizes the possibility of expert forecast estimates. Applying this approach in practice will make it possible to increase the efficiency of the foresight methods by reducing time costs, as well as the rational use of monetary and human resources. In this regard, there is a need for a qualitative examination of the results of scientific activity in order to increase its effectiveness. Furthermore, as the most effective tool for choosing priorities in the field of science and technology, the authors proposed the foresight methods. The mechanism for choosing priorities in forecasting research and expert forecast assessment is proposed. The determination of strategic research areas contributes to the competitiveness of scientific and technical activities. The results of this study can be the basis for further research works on improving the methodology for identifying the directions of development of science and technology.

Consideration of scientific concepts on the problem under consideration, comparison of the conclusions of the article with the conclusions of studies of other authors made it possible to summarize the results of the study. Thus, foresight-oriented methodologies used to evaluate scientific and technical activities will allow to provide a systemic dialogue between science and expert community.

The result of the research was the modernization of foresight methods, based on the application of algorithm for conducting qualitative expertise, which is aimed at the most efficient use of existing intellectual potential in order to solve urgent problems facing the state. The results obtained are the methodological basis for creating a system of integrated application of foresight methods for selecting priority areas of research.

There is a major limitation in this study that could be addressed in our research: the scaling requires additional financial resources.

We believe that our research is the basis for creating a regional research and production center in Kazakhstan. It contributes to the formation of such a scientific potential that would intensify the construction of innovative economies in the region. Results of the research can be developed to measure the overall level of scientific training and the conditions for its growth.

**Author Contributions:** Conceptualization, R.Z. and S.S.; Methodology—A.A.; investigation A.N.; writing—original draft preparation, A.N. All authors have read and agreed to the published version of the manuscript.

**Funding:** This study was funded and supported by the Science Committee of the Ministry of Education and Science of the Republic of Kazakhstan No AP05132160 "development and implementation of the foresight-oriented methods of educational work of doctoral students and undergraduates in the educational process."

**Conflicts of Interest:** The authors declare no conflict of interest.

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
