# Peer review of "The Use of the Foresight Methods in Developing an Algorithm for Conducting Qualitative Examination of the Research Activities Results on the Example of the Republic of Kazakhstan"

_mathematics, doi:10.3390/math8112024_

Round 1

Reviewer 1 Report

Dear author(s),
Writing a scientific article is a long journey and it is obvious you have just started it.
After reading your proposal, I have some recommendations for you in order to improve the quality of the manuscript.

The title of your proposal is: "The Use of the Foresight Methods in Developing an Algorithm for Conducting Qualitative Examination of the Research Activities Results". I have read your manuscript and I saw that you used data from Kazakhstan.
So, in order to be honest with your readers, I suggest to update the title by including a reference to the context.
Maybe something like "The Use of the ...... A Kazakhstan Case Study".

Page 5, Figure 1: The title of the figure is "Algorithm for conducting qualitative expertise in conducting foresight studies".
Please revise the title because the word "conducting" appears twice in the same title.

Section 4, page 4 is entitled "Results" and you present the figure 1. But the reader can't understand HOW the figure 1 was generated.
Please include a new section with previous researches and methodologies and then describe how you obtained your own methodology.

The Introduction section (pages 1-2) is pretty poor and your bibliographical references are a little bit deprecated. I consider you really must improve this section.
I recommend you to improve the general context of your research by citing some valuable resources in the Introduction of you manuscript: https://www.econstor.eu/handle/10419/168734 (the article can be downloaded for free from the bottom of the web site) and https://www.researchgate.net/profile/Octavian_Dospinescu/publication/341552612_perception_over_e-learning_tools_in_higher_education_comparative_study_romania_and_moldova/links/5ec6ba1692851c11a87d8cb4/perception-over-e-learning-tools-in-higher-education-comparative-study-romania-and-moldova.pdf and https://papers.ssrn.com/sol3/papers.cfm?abstract_id=3400671.
By including these references, you increase the general context of your research by presenting some relevant aspects from the educational-research field.

Page 6, equations 1, 2, 3: you use the Ki3 and Ki4 coefficients for the "personal qualities of the expert".
Please clearly describe what "personal qualities of the expert" mean, because I didn't find any description in the manuscript.
The reader must clearly understand the coefficients and their meanings.

Page 7, Figure 2: On the X Axis you have 2013r, ..., 2017r.
I don't understand what "r" letter means.
Please revise the chart or describe the meaning of this particle.
Also, in figure 2 you don't have the scale on the Y Axis.
You must revise the chart and include a scale on the Y Axis so that the reader know the values you use.

Page 9, Figure 3: The first item in the chart is "Agricultural and BILOLOGICAL sciences".
Please revise and correct the word "bilological".
You also must include percentages or values in chart, so that the user understand the general context.

Page 9, Figure 4: You present the number of Kazakhstani publications in Web of Science and Scopus.
But the figure contains only the colored series, without the scale.
So, please include the scale so that the readers know the values (numbers) of publications.
Also, the X axis must contain values and description. Are there the values from the last 5 years represented in the chart?
Please revise the chart, according to the research standards.

Page 10, table 3, second column: Please put a space between "availability" and "of a scientific leader".
The same situation on the last row of the table.

Page 10, table 3: You present two kinds of criteria: socio-economic and research criteria.
I don't understand the methodology you used to select this system of criteria.
As a recommendation, you should include a paragraph where you describe the steps and methodology used to generate this set of criteria.

Equations 5, 6, 7, 8 contains Russian letters (I'm not 100% sure, but they definitely are not English letters).
Taking into consideration that Mathematics is an prestigious international journal, you must change the notations and use only international symbols.
Also, you should invoke previous research studies in this area.

Page 13, section Discussion: You should include a section where you must validate your algorithm/model.
Now, your manuscript is just a proposal. But in the scientific world, every proposal must be tested and validated.
Put a short discussion about your algorithm proposal regarding other countries than Kazakhstan. Is the algorithm efficient in other contexts?!?

Pages 14-15, section Conclusions: Include a paragraph where you must present the limitations of your research.
Also, clearly describe the future directions of research.

Dear author(s), please consider all the above remarks as being constructive remarks in order to improve the general quality of your manuscript.

Kind Regards.

Author Response

We would like to express our gratitude for your contribution to our article «The use of the foresight methods in developing an algorithm for conducting qualitative examination of the research activities results on the example of the Republic of Kazakhstan».

 We have tried to improve the quality of the manuscript according to your recommendations:

The title of your proposal is: "The Use of the Foresight Methods in Developing an Algorithm for Conducting Qualitative Examination of the Research Activities Results". I have read your manuscript and I saw that you used data from Kazakhstan. So, in order to be honest with your readers, I suggest to update the title by including a reference to the context. Maybe something like "The Use of the ...... A Kazakhstan Case Study".

The use of the foresight methods in developing an algorithm for conducting qualitative examination of the research activities results on the example of the Republic of Kazakhstan.

Page 5, Figure 1: The title of the figure is "Algorithm for conducting qualitative expertise in conducting foresight studies". Please revise the title because the word "conducting" appears twice in the same title.

Figure 1. Algorithm for qualitative expertise in conducting foresight studies

Section 4, page 4 is entitled "Results" and you present the figure 1. But the reader can't understand HOW the figure 1 was generated. Please include a new section with previous researches and methodologies and then describe how you obtained your own methodology.

Methodologies

Methods of bibliometry, scientifometry, patent analysis, method of observation and collection of facts, method of modeling were used.

Bibliometric analysis, or the method of calculating the number of publications, involves a quantitative assessment of the document flow and a quantitative analysis of scientific documents from different fields of knowledge. The fields of science and individual sections, which occupy a leading place in the structure of scientific knowledge according the number of scientific publications. distinguished on the results of bibliometric analysis.

Science measurement, or citation analysis, is carried out by studying bibliographic references in publications of scientific periodicals databases (Web of Science, SCOPUS, TR) in order to identify the citation of publications, forming a certain direction of science, the number of references (self-citation is excluded), the total number publications by direction.

Patent analysis uses statistical methods for processing patent information arrays including the analysis the dynamics of inventive activity’s curves for each scientific and technical direction; The result of analyses includes the construction of cumulative patent series characterized by increase of total number of patents related to this area. Our methodology allows to cover not only research criteria, but also socio-economic, such as risk assessment, the presence of a potential investor, applicability, market access prospects and financing.

It is necessary to specify the systemic approach in the study and in raising the question.

The adequacy of the application of the forsite methodology is due to its advantages over other methods of expert assessments, giving the following possibilities:

- evaluating not only quantitative but also qualitative information;

- determining the weight coefficients of objects, processes and phenomena that cannot be quantified in the metric system of measures;

- recreating an objective picture of experts’ opinions on the problem being studied, which excludes the averaging of their assessments (minimization of expert subjectivity);

- establishing a measure of contradictory judgments of each expert and thus determine the degree of reliability, trust in the obtained result.

The Introduction section (pages 1-2) is pretty poor and your bibliographical references are a little bit deprecated. I consider you really must improve this section. I recommend you to improve the general context of your research by citing some valuable resources in the Introduction of you manuscript: https://www.econstor.eu/handle/10419/168734 (the article can be downloaded for free from the bottom of the web site) and https://www.researchgate.net/profile/Octavian_Dospinescu/publication/341552612_perception_over_e-learning_tools_in_higher_education_comparative_study_romania_and_moldova/links/5ec6ba1692851c11a87d8cb4/perception-over-e-learning-tools-in-higher-education-comparative-study-romania-and-moldova.pdf and https://papers.ssrn.com/sol3/papers.cfm?abstract_id=3400671. By including these references, you increase the general context of your research by presenting some relevant aspects from the educational-research field.

Scientists around the world are researching students' perceptions of e-learning platforms and tools. Recent research has shown that the use of electronic technologies and platforms is aimed at improving the educational process. Currently, the authors (Octavian Dosnescu, Nicoleta Dospinescu) find the field of e-learning as the intersection of commercial, educational and technological interests [2].

  1. Maksimtseva, I.A. Fundamentals of a knowledge-based economy. Moscow: Creative Economy Publishing House, 2010.
  2. Octavian Dosnescu, Nicoleta Dospinescu Perception over E-learning tools in higher education: comparative study Romania and Moldova. Proceedings of the IE 2020 International Conference. https://www.researchgate.net/publication/341552612
  3. Brumer, V.; Konnola, T.; Salo, A. Diversity in Foresight Research. The practice of selecting innovative ideas.   T. 4. 2011, No. 4, 56-68. http://ecsocman.hse.ru/data/2011/11/28/1270192366/56-68.pdf
  4. Tapinos, E.; Pyper, N. Forward looking analysis: Investigating how individuals do’ foresight and make sense of the future. Technological Forecasting and Social Change, 2018, 126, 292-302. https://doi.org/10.1016/j.techfore.2017.04.025
  5. Shelyubskaya, N.V. Foresight is a mechanism for determining the priorities of the formation of a knowledge society in Western Europe. Kiev: Fenix, 2007.
  6. Danova, M.A. Priority selection technique for predicting the scientific and technological development of large-scale facilities based on the Foresight technology. Aerospace Engineering and Technology, 2013, 7 (104), 227-231.
  7. Daniel Homocianu, Napaleon-Alexandru Sireteanu, Octavian Dospinescu, Dinu Airinei. An Analysis of Scientific Publications on 'Decision Support Systems' and 'Business Intelligence' Regarding Related Concepts Using Natural Language Processing Tools.  Proceedings of the IE. 2019. International Conference, Bucharest, pp.99-104. https://papers.ssrn.com/sol3/papers.cfm?abstract_id=3400671.

Page 6, equations 1, 2, 3: you use the Ki3 and Ki4 coefficients for the "personal qualities of the expert". Please clearly describe what "personal qualities of the expert" mean, because I didn't find any description in the manuscript. The reader must clearly understand the coefficients and their meanings.

whereis the coefficient reflecting the self-esteem of the expert on his j personal quality; is the number of personal qualities of the expert (characterized by his behavior and relations with the verifiable)

Page 7, Figure 2: On the X Axis you have 2013r, ..., 2017r. I don't understand what "r" letter means. Please revise the chart or describe the meaning of this particle. Also, in figure 2 you don't have the scale on the Y Axis. You must revise the chart and include a scale on the Y Axis so that the reader know the values you use.

Page 9, Figure 3: The first item in the chart is "Agricultural and BILOLOGICAL sciences". Please revise and correct the word "bilological". You also must include percentages or values in chart, so that the user understand the general context.

Page 9, Figure 4: You present the number of Kazakhstani publications in Web of Science and Scopus. But the figure contains only the colored series, without the scale. So, please include the scale so that the readers know the values (numbers) of publications. Also, the X axis must contain values and description. Are there the values from the last 5 years represented in the chart? Please revise the chart, according to the research standards.

Page 10, table 3, second column: Please put a space between "availability" and "of a scientific leader". The same situation on the last row of the table.

Table 3. System of criteria for assessing research areas

Research areas assessment criteria

socio-economic criteria:

research criteria:

- risk assessment;

- availability of a potential investor;

- applicability;

- prospects for entering the market;

- financing.

- availability of a scientific leader;

- availability of production capacity;

- availability of scientific specialists;

- effectiveness of research activities;

- availabilityof an experimental production base.

Page 10, table 3: You present two kinds of criteria: socio-economic and research criteria. I don't understand the methodology you used to select this system of criteria. As a recommendation, you should include a paragraph where you describe the steps and methodology used to generate this set of criteria.

Currently, the governments with developed market economies are pursuing policies of creating a so-called "new economy" or "knowledge-based economy." The active formation of the new economy has been carried out over the past twenty years with global information networks, communications and the Internet. The strategic development of the national economy of various countries, including Kazakhstan, is primarily inextricably linked with the formation of new economic relations. The authors understand the economics of knowledge to be  the system of socio-economic relations of the innovative-digital type, based on the priority of intellectual human capital, the intensive development of digital technologies, knowledge-intensive production, high-technology industries and continuing professional  education.

Equations 5, 6, 7, 8 contains Russian letters (I'm not 100% sure, but they definitely are not English letters). Taking into consideration that Mathematics is an prestigious international journal, you must change the notations and use only international symbols. Also, you should invoke previous research studies in this area.

Quantitative criteria (expressed by quantitative values) include - financing; availability of scientific specialists; effectiveness of research activities; availability of production facilities and availability of an experimental and production base.

To find the values of the criterion "Financing" (), the share of capital costs in the internal costs of research, development and purchase of equipment is determined [38].

(5)

where Zres – research and development capital expenditures; Zcap – capital and current expenses for the purchase of equipment; Zintern– internal research and development costs.

The criterion of «Availability of scientific specialists» (Ks) is  the specific gravity of the number of highly qualified specialists in the total number of scientists [41]:

                                      (6)

where Кqs – the number of highly qualified specialists; Кtot – the total number of scientists.

Specialists of the higher classification include doctors and candidates of sciences, and their numbers are calculated taking into account weight coefficients of 0.8 and 0.5 for doctors and candidates of sciences, respectively:

                                     (7)

Criteria “Availability of an experimental and production base” (Кex) and “Availability of production facilities” (Кpf) are determined by analyzing accounting and statistical reporting documents (information on the availability of production capacities, balance sheet, balance of production capacities, etc.) of a university of a particular research area. These two criteria take the value 1 if the availability is confirmed and 0 if this is not the case.

The criterion of «Research Activity Efficiency» (Кrae) implies a total calculation of development indicators for a particular research area, such as - the total number of valid university patents (Кvup); the number of filed patent applications and applications for utility models (for the last year) (Кpat); number of acquired rights to patents and patent licenses (Кrpat): 

Page 13, section Discussion: You should include a section where you must validate your algorithm/model. Now, your manuscript is just a proposal. But in the scientific world, every proposal must be tested and validated. Put a short discussion about your algorithm proposal regarding other countries than Kazakhstan. Is the algorithm efficient in other contexts?!?

The project team piloted the development of Forsite information technology in the educational process of undergraduates and doctoral students, and worked out administrative procedures involving experts and students. The web application (information system) was introduced in the NAO "Almaty University of Energy and Communications". We consider our algorithm proposal to be efficient in the Commonwealth of Independent States (CIS) and in European Union (EU).  On an international scale, the project is a unique experiment in the field of institutional economy, aimed at creating an institution that identifies the most promising scientific directions in the self-organization mode.

Pages 14-15, section Conclusions: Include a paragraph where you must present the limitations of your research. Also, clearly describe the future directions of research.

There is a major limitation in this study that could be addressed in future research: the following scaling requires additional financial resources .

We believe that there is a need to create a regional research and production center in Kazakhstan that contributes to the formation of such a scientific potential that would intensify the construction of innovative economies in the region. Results of the research can be developed to measure the overall level of scientific training and the conditions for its growth.

As all corrections have been made, we hope that the manuscript will be accepted.

Zhanbayev Rinat,

Ph.D. of  Economics sciences,

member of the National Scientific Council of the Republic of Kazakhstan

in the field of "Information, telecommunication and space

technologies, scientific research in the field of natural sciences"

Reviewer 2 Report

The paper, concerned with the increasing of the efficiency of the foresight methods, aims at ensuring a systemic relationship between different sectors of economy and higher education in research area. The authors have developed an algorithm based on a systematic approach regarding the foresight methods along with the utilization of the methods concerning bibliometrics, scientometrics, patent analysis and forecasting.

A potential benefit of the paper would be to provide a new perspective to improve the efficiency of the foresight methods.

One of my suggestions is to include more recent works in the reference.

Another suggestion would be to improve the graphic visual aid, if possible.

Also a comparative analysis could be provided to make the study's contribution more evident as the motivation part.

If possible, the structure of the paper can be revised.

Yours sincerely

Author Response

(The authors gave the same response as above.)

Round 2

Reviewer 1 Report

Dear author(s),

I have read the new version of the paper and I saw that some of my previous remarks were not fully addressed. Also, some new issues appear in the present version.

Please find below my remarks about the new version of your manuscript.

  1. Page 5, section "Methodologies" should be renamed "Methodology".
  2. Page 5, in the section "Methodologies" you say "The adequacy of the application of the forsite methodology...". Please revise the word "forsite".
  3. Page 7, Figure 2: On the X Axis you have 2013r, ..., 2017r. I don't understand what "r" letter means. Please revise the chart or describe the meaning of this particle. Also, in figure 2 you don't have the scale on the Y Axis. You must revise the chart and include a scale on the Y Axis so that the reader know the values you use.

  4. Page 9, Figure 3: The first item in the chart is "Agricultural and BILOLOGICAL sciences". Please revise and correct the word "bilological". You also must include percentages or values in chart, so that the user understand the general context.

  5. Page 10, Figure 4: You present the number of Kazakhstani publications in Web of Science and Scopus. But the figure contains only the colored series, without the scale. So, please include the scale so that the readers know the values (numbers) of publications. Also, the X axis must contain values and description. Are there the values from the last 5 years represented in the chart? Please revise the chart, according to the research standards.

  6. At page 14, in the bottom of the page, you say: "This study complements the theory and practice of foresight as a forecasting technology, namely:". Then you have a list. Please put some bullets before every item of the enumeration.
  7. In the Conclusion section, you say: "There is a major limitation in this study that could be addressed in future research: the following scaling requires additional financial resources." Regarding the limitations, in a scientific article, the authors must present the actual limitations of the research. So, I recommend to reformulate this sentence.
  8. Regarding the references list, the name of the first author for the reference 2 is wrong. Please revise it. And also correct it in the first paragraph at page 2. Instead of Dosnescu, the correct name is Dospinescu. Also, for the reference #5, please specify the country: Ukraine.
  9. In the Conclusion section, you say: "We believe that there is a need to create a regional research and production center in Kazakhstan that contributes to the formation of such a scientific potential that would intensify the construction of innovative economies in the region." This sentence has no argument based on your research. It seems to be your subjective perception and not a scientific result. So please, revise and reformulate this sentence.

Dear author(s),

Please consider all the above suggestions as being constructive remarks in order to improve the quality of your manuscript.

Kind Regards!

Author Response

We are  thankful to the reviewers of our article «The use of the foresight methods in developing an algorithm for conducting qualitative examination of the research activities results on the example of the Republic of Kazakhstan» for the time they have taken to give us valuable input to the article.

We have made all the required corrections, despite they do not express criticism and have just the recommendatory character.

We kindly ask you to accept our article for publishing, hoping the manuscript doesn’t require further changes.

Zhanbayev Rinat

Round 3

Reviewer 1 Report

Dear author(s),

After reading the new version of your manuscript proposal, I consider that you addressed all my remarks and recommendations from the previous rounds of review.

Kind Regards!